# Chemical Constituents with Inhibitory Activity of NO Production from a Wild Edible Mushroom, *Russula vinosa* Lindbl, May Be Its Nutritional Ingredients

**DOI:** 10.3390/molecules24071305

**Published:** 2019-04-03

**Authors:** Guodong Zhang, Huawei Geng, Chunxia Zhao, Fangyi Li, Zhen-Fa Li, Boshu Lun, Chunhua Wang, Heshui Yu, Songtao Bie, Zheng Li

**Affiliations:** 1College of Pharmaceutical Engineering of Traditional Chinese Medicine, Tianjin University of Traditional Chinese Medicine, Tianjin 300193, China; pharmgd@126.com (G.Z.); zhaochunxia199411@163.com (C.Z.); li_fangyi@hotmail.com (F.L.); lizhenxiaofa@163.com (Z.-F.L.); lbsxcl@163.com (B.L.); hs_yu08@163.com (H.Y.); biesongtao@126.com (S.B.); 2Chongqing Municipal Public Security Bureau Evidence Identification Center, Chongqing 400707, China; huaweigeng0414@sina.com; 3Tianjin Key Laboratory of Modern Chinese Medicine, Tianjin University of Traditional Chinese Medicine, Tianjin 300193, China

**Keywords:** *Russula vinosa* Lindbl, chemical constituents, isolation and identification, cytotoxic, NO production

## Abstract

*Russula vinosa* Lindbl is a wild edible mushroom that is usually used for original material of food and soup and has rich nutritional value. What are the nutritional ingredients? In order to answer this question, we investigated the chemical constituents of this wild functional food. Six new compounds (**1**–**6**), together with nine known ones (**7**–**15**), were isolated from *R. vinosa*. The six new compounds were named as vinosane (**1**), rulepidadione C (**2**), (24*E*)-3,4-seco-cucurbita-4,24-diene-26,29-dioic acid-3-methyl ester (**3**), (24*E*)-3,4-seco-cucurbita-4,24-diene-26-oic acid-3-ethyl ester (**4**), (24*E*)-3β-hydroxycucurbita-5,24-diene-26,29-dioic acid (**5**), and (2*S*,3*S*,4*R*,2′*R*)-2-(2′-hydroxydocosanoylamino)eicosane-1,3,4-triol (**6**). Their structures were determined based on spectroscopic methods including HR-ESI-MS, 1D, and 2D NMR. Moreover, a cell counting kit-8 (CCK-8 kit) was used to screen for the cytotoxicity of compounds **1**–**5** and **7**–**13** on mouse macrophage RAW 264.7 cells. The results showed that compounds **1**–**5** and **7**–**13** had no obvious cytotoxicity. In addition, the inhibitory effects on nitric oxide (NO) production in lipopolysaccharide (LPS)-activated mouse macrophage RAW 264.7 cells were evaluated. Compounds **1**, **3**, **4**, **7**, **12**, and **13** showed moderate inhibitory activity on NO production.

## 1. Introduction

*Russula vinosa* Lindbl belongs to the genus of *Russula*, a multifunctional edible fungus, which is distributed mainly in Northwest Fujian province and Jiangxi province in China. Studies have shown that *R. vinosa* has good antitumor and anti-inflammatory activities. Related research has revealed that the water-soluble components of *R. vinosa* are mainly polysaccharides, and the fat-soluble components are sesquiterpenes, triterpenes, steroids, ceramides, fatty acids, and other compounds [1,2,3]. Research progress of fat-soluble components in edible mushroom has shown that they have several bioactivities, such as anti-tumor, anti-inflammation, anti-oxidant, and antibacterial effects [4,5,6,7]. Consequently, the chemical analysis of the *Russula* edible fungus was performed by the natural product chemistry and food chemistry scientists. The chemical constituents of several other mushrooms have been investigated, such as *R. japonica*, *R. subnigricans*, *R. lepida*, *R. cyanoxantha*, etc. [8,9,10,11]. *R. vinosa* has been used as a multifunctional edible food in our life and although there are some studies on the water-soluble polysaccharides, few systematic investigations on the fat-soluble components have been performed. In order to systematically investigate the fat-soluble components of *R. vinosa*, we separated and identified the fat-soluble components of *R. vinosa* by silica gel column, ODS column, gel column, and preparative high-performance liquid chromatography (Pr-HPLC), and finally 15 compounds (Figure 1) were determined according to their physicochemical data and spectral data such as 1D, 2D-NMR, and MS. Interestingly, six new compounds were found in this fungus, and also the inhibitory effect of those compounds on nitric oxide (NO) production in lipopolysaccharide-activated macrophage RAW 264.7 cells was evaluated. As a result, compounds **1**, **3**, **4**, **7**, **12,** and **13** showed moderate inhibitory activity on NO production at higher concentrations. 

## 2. Results and Discussion

Fifteen compounds (Figure 1) were determined according to their physicochemical data and spectral data such as 1D, 2D-NMR, and MS. The compounds included four sesquiterpenes: vinosane (**1**), rulepidadione C (**2**), 7α,8α,13-trihydroxy-marasm-5-oic acid γ-lactone (**12**), and aristolone (**13**); eight triterpenes: (24*E*)-3,4-seco-cucurbita-4,24-diene-26,29-dioic acid-3-methyl ester (**3**), (24*E*)-3,4-seco-cucurbita-4,24-diene-26-oic acid-3-ethyl ester (**4**), (24*E*)-3β-hydroxycucurbita-5,24-diene-26,29-dioic acid (**5**), (24*E*)-3,4-secocucurbita-4,24-diene-3,26,29-trioic acid (**7**), (24*E*)-3,4-secocucurbita-4,24-diene-3,26-dioic acid (**8**), (24*E*)-3β-hydroxycucurbita-5,24-diene-26-oic acid (**9**), rosacea acid B (**10**), and rosacea acid A (**11**); together with three compounds containing N element as follows: (2*S*,3*S*,4*R*,2′*R*)-2-(2′-hydroxydocosanoylamino)eicosane-1,3,4-triol (**6**), 7,8-dimethylalloxazine (**14**), and l-pyroglutamic acid (**15**). Among these compounds, compounds **1**–**6** were new compounds, and **14** and **15** were isolated from this genus for the first time. In this research, we also investigated the cytotoxic activity of compounds **1**–**5** and **7**–**13** using a CCK-8 kit assay. More than that, the inhibitory effect of those compounds on NO production in lipopolysaccharide-activated macrophage RAW 264.7 cells was also evaluated. As a result, all test compounds showed little cytotoxic activity on macrophage RAW 264.7 cells, and compounds **1**, **3**, **4**, **7**, **12**, and **13** showed moderate inhibitory activity on NO production at higher concentrations.

### 2.1. Structural Elucidation

Chromatography of the 95% ethanol extract from the *R. vinosa* over silica gel, MCI gel, Sephadex LH-20, and Pr-HPLC with different solvent systems afforded compounds **1**−**15**. The known compounds **7**–**15** were identified by comparison of their experimental and spectral data with literature data [12,13,14,15,16,17].

Compound **1** was obtained as colorless needle crystal with the molecular formula of C_15_H_22_O_3_ based on the protonated molecular ion at *m/z* 249.1503 [M − H]^−^ (calcd for: 249.1491), implying 5 degrees of unsaturation. The ^1^H-NMR spectrum of compound **1** (Table 1) showed the presence of two methyls at δ 0.96 (3H, d, *J* = 7.2 Hz) and *δ* 0.90 (3H, d, *J* = 6.8 Hz), a methylene linked to –O– at δ 4.78 (2H, m), and a double bond at δ 6.82 (1H, s). Fifteen carbon signals were shown from the ^13^C-NMR (Table 2) (CD_3_OD, 100 MHz) spectrum, which ascribed to two methyls, five methylenes, five methines, and three quaternary carbons in combination with the DEPT-135 spectrum. δ 168.0 (C-14) showed a carbonyl signal; combined with the signals of δ 129.5 (C-2) and δ 152.3 (C-3), the structure of α,β-unsaturated carboxylic acid was deduced. In addition, there was a signal of quaternary carbon at δ 92.6 (C-4). The ^1^H-^1^H COSY data of **1** showed the correlations (Figure 2) between δ 6.82 (H-2) and δ 4.78 (H-1), δ 0.90 (H-15), and δ 1.75 (H-7), and δ 0.96 (H-13) and δ 2.13 (H-11), which could determine the connection mode of carboxyl group and the substituted position of two methyl groups. The correlations between the δ 2.12 (H-11) and δ 1.86 (H-12) and δ 1.48 (H-10); the δ 1.48 (H-9) and δ 2.05 (H-8) and δ 1.48 (H-10); the δ 1.75 (H-7) and δ 2.05 (H-8) and δ 1.53 (H-6); the δ 1.53 (H-6) and δ 1.34 (H-5) and δ 1.75 (H-7); and the δ 1.86 (H-12) and δ 2.05 (H-8) revealed an “octahydro-1H-indene” fragment. The HMBC data showed correlations (Figure 2) between δ 6.82 (H-2) and δ 72.2 (C-1), δ 92.6 (C-4), and δ 168.0 (C-14), which can determine the structure of α,β-unsaturated carboxylic acid and the furan ring. Since δ 2.05 (H-8), δ 1.56 (H-5), and δ 1.28 (H-6) were all correlated with δ 92.6 (C-4) and the unsaturation degree of this compound, we can speculate that A and B were connected to form a spiro structure through C-4 (Figure 2). The relative configuration can be determined by the NOESY spectrum. Thus, the structure of this compound can be determined and named as vinosane. 

Compound **2** was obtained as white amorphous powder with the molecular formula of C_15_H_22_O_3_ based on the protonated molecular ion at *m/z* 251.1651 [M + H]^+^ (calcd for: 251.1647), implying 5 degrees of unsaturation. The ^1^H-NMR spectrum (Table 1) of compound **2** displayed signals of three methyls at δ 0.88 (3H, s), δ 1.15 (3H, d, *J* = 6.8 Hz), and δ 1.42 (3H, s). In combination with the ^13^C-NMR spectrum, fifteen carbon signals were shown to be a sesquiterpene. The signals of δ 211.2 (C-8) and δ 211.8 (C-1) indicated that there were two ketone carbonyl groups. A signal of carbon linked to hydroxyl group at δ 72.0. The DEPT-135 and ^13^C-NMR data (Table 2) ascribed to three methyls, four methylenes, four methines, and four quaternary carbons. The HSQC data showed a correlation between δ 72.0 (C-12) and δ 3.37 (2H, m). It showed that there was a methylol group in the structure. Compared with the literature [16], this compound was deduced as an aristolane-type sesquiterpene. The ^1^H-^1^H COSY data of compound **2** showed correlations (Figure 2) between the δ 2.27 (H-4) and δ 1.15 (H-15); the δ 1.75 (H-7) and δ 1.49 (H-6); the δ 3.08 (H-10) and δ 2.16 (H-9α) and δ 2.49 (H-9β); and the δ 2.33 (H-2) and δ 1.75 (H-3α) and δ 2.02 (H-3β), respectively. The HMBC data showed correlations (Figure 2) between the δ 3.08 (H-10), δ 2.33 (H-2), and δ 211.8 (C-1); the δ 1.49 (H-6), δ 1.75 (H-7), and δ 2.16 (H-9); and δ 211.2 (C-8), the δ 0.88 (H-14) and δ 35.4(C-6), δ 41.0(C-5), and δ 51.4 (C-10); the δ 1.15 (H-15) and δ 32.0 (C-3) and δ 41.0 (C-5); the δ 1.42 (H-13) and δ 72.0 (C-12) and δ 33.8 (C-7, C-11); the δ 3.37 (H-12) and δ 13.8 (C-13) and δ 33.8 (C-11); the δ 1.49 (H-6), δ 1.75 (H-7), and δ 72.0 (C-12); the δ 3.88 (H-10) and δ 1.42 (H-13), δ 1.49 (H-6),and δ 1.75 (H-7); and the δ 1.49 (H-6) and δ 0.88 (H-14) and δ 1.15 (H-15), respectively. The relative configuration was determined by the NOESY spectrum (Figure 2). So, the structure of compound **2** can be determined by the above analysis and named as rulepidadione C. 

Compound **3** was obtained as white amorphous powder. Its molecular formula was deduced as C_31_H_48_O_6_ by a HR-ESI-MS pronated molecular ion at *m/z* 515.3375 [M − H]^−^ (calcd for: 515.3373), corresponding to eight degrees of unsaturation. The ^1^H-NMR spectrum of compound **3** (Table 1) showed the presence of seven methyls at δ 0.66 (3H, s), δ 0.82 (3H, s), δ 0.91 (3H, d, *J* = 6.0 Hz), δ 1.19 (3H, s), δ 1.84 (3H, s), δ 1.84 (3H, s), and δ 3.66 (3H, s), along with an olefinic hydrogen at δ 6.90 (1H, m). Thirty-one carbon signals were shown from the ^13^C-NMR spectrum (Table 2), which ascribed to seven methyls, ten methylenes, five methines, and nine quaternary carbons in combination with DEPT-135 data. The signals of δ 121.9 (C-4), δ 126.8 (C-25), δ 146.0 (C-24), and δ 154.7 (C-5) indicated the existence of two groups of double bonds, among them, δ 146.0 (C-24) was a methylene and others were quaternary carbons. Furthermore, the δ 173.8 (C-26), δ 174.4 (C-3), and δ 175.8 (C-29) suggested three carbonyls. The signals of δ 3.66 (3H, s) and δ 51.7 (C-31) were methoxyl signals. The HMBC data showed the correlations (Figure 2) between the δ 1.84 (H-27), δ 6.99 (H-24), and δ 173.8 (C-26); and the δ 1.84 (H-28) and δ 121.9 (C-4), δ 154.7 (C-5), and δ 175.8 (C-29). The signals of δ 3.66 (H-31), δ 2.31 (H-2), and δ 1.68 (H-1) were correlated with δ 174.4 (C-3), which indicated that there was a carboxylic acid methyl ester, thus the planar structure of the compound was determined. The NOESY spectrum showed that there were correlations (Figure 2) between δ 1.19 (H-19) and δ 0.82 (H-18), and δ 2.48 (H-8), which conformed to the same relative configuration as in the literature [16]. Thus, compound **3** was identified as (24*E*)-3,4-seco-cucurbita-4,24-diene-26,29-dioic acid-3-methyl ester. 

Compound **4** was obtained as white amorphous powder. Its molecular formula was deduced as C_32_H_52_O_4_ by an HR-ESI-MS pronated molecular ion at *m/z* 523.3405 [M + Na]^+^ (calcd for: 523.3763), corresponding to seven degrees of unsaturation. The ^1^H-NMR spectrum of compound **4** (Table 1) showed the presence of eight methyls at δ 0.63 (3H,s), δ 0.81 (3H,s), δ 0.91 (3H,d, *J* = 6.1 Hz), δ 1.14 (3H,s), δ 1.25 (3H, t, *J* = 7.1 Hz), δ 1.54 (3H,s), δ 1.64 (3H,s), δ 1.83 (3H, s), an olefinic hydrogen at δ 6.90 (1H, s), and an –O–CH_2_– at δ 4.13 (2H, s). The ^13^C-NMR spectrum (Table 2) showed the presence of thirty-two carbon signals, which ascribed to eight methyls, eleven methylenes, five methines, and eight quaternary carbons in combination with DEPT-135 spectrum. The signals of δ 123.2 (C-4), δ 126.7 (C-25), δ 133.8 (C-5), and δ 146.0 (C-24) indicated the existence of two groups of double bonds, and δ 173.2 (C-26) and δ 174.7 (C-3) suggested the presence of two carbonyls. Comparing the NMR data of compound **4** with the literature [18], we could see that these two structures had the same skeleton. The difference between the two structures is that compound **4** had one more –O–CH_2_– and one more methyl and lost one carbonyl signal. The HMBC data showed remote correlations (Figure 2) between the δ 4.13 (H-31) and δ 14.4 (C-32) and 174.7 (C-3), and the δ 2.25 (H-2) and δ 174.7 (C-3), which inferred that carboxylic acid formed ethyl carboxylate. The other correlations (Figure 2) were observed for the *δ* 1.83 (H-27) with δ 126.7 (C-25), δ 46.0 (C-24), and δ 173.2 (C-26); for the δ 6.90 (H-24) with δ 173.2 (C-26); for the δ 2.12 (H-23) with δ 146.0 (C-24); for the δ 1.54 (H-29) and δ 1.64 (H-28) with δ 123.2 (C-4) and δ 133.8 (C-5), respectively. In summary, the planar structure of compound **4** could be determined. The NOESY spectrum showed the correlations (Figure 2) between the δ 1.14 (H-19) and *δ* 0.81 (H-18), and *δ* 1.93 (H-8); and the *δ* 2.42 (H-10) and *δ* 0.63 (H-30), which conformed the same relative configuration as in the literature [16]. Thus, the chemical structure of compound **4** was characterized as (24*E*)-3,4-seco-cucurbita-4,24-diene-26-oic acid-3-methyl ester.

Compound **5**, a white amorphous powder, gave a molecular formula of C_30_H_46_O_5_ ([M − H]^−^, calcd for C_30_H_46_O_5,_ 485.3267) with 8 degrees of unsaturation, based on its HR-ESI-MS. Six groups of methyl signals were detected in ^1^H-NMR and δ 4.23 (1H, m), suggesting an oxygen substitution. Two signals of olefinic hydrogens were showed at δ 6.01 (1H, d, *J* = 5.6 Hz), δ 7.23 (1H, m). Thirty carbon signals were showed in the ^13^C-NMR data, which ascribed to six methyls, nine methylenes, seven methines, and eight quaternary carbons in combination with DEPT-135 data. The ^13^C-NMR showed the presence of two carboxyl substitutes at δ 171.0 (C-26) and δ 180.5 (C-29); two groups of olefinic bonds at δ 123.3 (C-6), δ 129.3 (C-25), δ 138.8 (C-5), and δ 142.9 (C-24); and a hydroxyl linked carbon at δ 72.9 (C-3). Comparing the above data with the literature [18], a cucurbitane-type triterpene was deduced. The position of substituents was determined by HMBC spectrum. From the correlations (Figure 2) between δ 1.66 (H-28) and δ 55.0 (C-4), δ 72.9 (C-3), δ 138.8 (C-5), and δ 180.5 (C-29), we can confirm that the 28-position methyl and 29-position carboxyl were linked to 4-position carbon. The other correlations (Figure 2) between δ 2.11 (H-27) and δ 129.3 (C-25), δ 142.9 (C-24), and δ 171.0 (C-26) can determine the position of another double bond and carboxyl group. Therefore, the chemical structure of the compound can be determined by the above analysis. The NOESY spectrum showed correlations (Figure 2) between the δ 0.97 (H-19) and δ 0.81 (H-18) δ 1.67 (H-8); the δ 2.44 (H-10) and δ 0.90 (H-30) and δ 1.66 (H-28); and the δ 1.66 (H-28) and δ 4.23 (H-3), which conformed to its relative configuration of compound **5** compared with the literature [16]. Thus, the chemical structure of compound **5** was characterized as (24*E*)-3β-Hydroxycucurbita-5,24-diene-26,29-dioic acid.

Compound **6**, a white granular powder, gave a molecular formula of C_42_H_85_NO_5_ ([M + H]^+^, calcd for C_42_H_85_NO_5,_ 684.6506) with one degree of unsaturation based on its LC-IT-TOF-MS. The ^1^H-NMR and ^13^C-NMR spectra of compound **6** (Table 3) displayed many overlapped hydrogen signals at *δ* 1.20–1.40 and many overlapped carbon signals at δ 30.3–30.5, respectively. Therefore, we could speculate that the compound contained an aliphatic chain. The ^1^H-NMR spectrum showed the presence of a reactive hydrogen at δ 8.61, and the signal of δ 4.31 (H-4), δ 4.38 (H-1), δ 4.45 (H-2), δ 4.53 (H-1), and δ 4.64 (H-2′) showed that there were multiple hydroxyl substituents. It can be seen that there were four carbons connected to hydroxyl groups in the structure at δ 62.4 (C-1), δ 77.2 (C-3), δ 73.4 (C-4), δ 72.9 (C-2′). The DEPT-135 data suggested that *δ* 62.4 (C-1) was a carbon signal of methylene, which linked to a hydroxyl group. The signal of δ 175.6 (C-1′) speculated that there was a carbonyl group. Consequently, the compound was inferred as a ceramide by comparison with the literature [19], whose core structure was shown in Figure 1(A). R_1_ and R_2_ were replaced by long aliphatic chains, and the length of aliphatic chains on both sides was determined by mass spectrometry. ESI-MS gives a fragment of *m/z* 339.35 in negative ion mode, The molecular formula was supposed to be the fragment after the amide bond disrupted, which was C_22_H_43_O_2_^−^ (calcd for: 339.3269) based on ESI-MS, so R_1_ is C_20_H_41_. Besides, HR-ESI-MS gave the fragment peak of *m/z* 344.2797 in negative ion mode (calcd for: 344.3165). According to the molecular weight, the other side of the aliphatic chain R_2_ was deduced to be C_16_H_33_. The NOESY data showed correlations between the δ 4.38 (H-1) and δ 5.14 (H-3) and δ 4.64 (H-2′); and the δ 4.45 (H-2) and δ 5.14 (H-3), which conformed to the relative configuration. Therefore, compound **6** was determined as (2*S*,3*S*,4*R*,2′*R*)-2-(2′-hydroxydocosanoylamino)eicosane-1,3,4-triol.

### 2.2. Marked Peaks of Isolated Compounds

After the isolation, 15 isolates were marked on the UPLC-Q/TOF-MS chromatograms in Figure 3. The identification of these 15 compounds could be seen in Appendix A.

### 2.3. Bioactivity Evaluation

In this study, the inhibition of NO production by LPS-induced RAW 264.7 mouse macrophages was measured. Twelve components were used to screen the potential inhibitory activity on NO production.

#### 2.3.1. Cytotoxic Activity Assay

The results (Figure 4) showed that there was no significant difference in cell viability between LPS, dexamethasone sodium phosphate (Dex), the monomer groups (50, 25, 12.5 μg/mL), and the control group. 

#### 2.3.2. Inhibitory Activity on NO Production Assay

In the results (Figure 5), compared with the control group, the release of NO in RAW264.7 cells was significantly increased after LPS stimulation, and the monomers inhibited the release of NO in different degrees. Compounds **1**, **3**, **4**, **7**, **12**, and **13** showed better activity at higher concentrations. Through this experiment, we screened some compounds with inhibitory activity on NO production.

## 3. Materials and Methods

### 3.1. General

HPLC was run on Agilent 1260 HPLC (Agilent, Palo Alto, CA, USA). Semi-preparative HPLC was performed on Waters 2489 equipped with a diode array detector and a C_18_ column (250 mm × 10 mm, 5 μm, Waters, Maple St. Milford, MA, USA). NMR spectra was measured on an AV-400 spectrometer (Bruker Corporation, Faellanden, Switzerland). Thin-layer chromatography (TLC) was performed on glass precoated silica gel GF_254_ plates (Qingdao Haiyang Chemical Co., Ltd, Qingdao, China), detection under UV light or by heating after spraying with 10% sulfuric acid (H_2_SO_4_) in 90% ethanol (EtOH). Column chromatography was performed on silica gel (200–300 mesh, Qingdao Marine Chemical Inc., Qingdao, China), and Sephadex LH-20 (Amersham Pharmacia Biotech, Uppsala, Sweden) were used for the chromatography column. Other chemicals and reagents of analytical grade were from Tianjin Concord Technology (Tianjin, China).

### 3.2. UPLC-QTOF-MS/MS Conditions

The chromatography was performed with a Waters Acquity UPLC BEH C_18_ column (2.1 × 100 mm, 1.7 µm; Waters, Milford, MA, USA) and the column temperature was maintained at 50 °C. One μL of sample was used for separation. The parameters of the mass spectrometer were set as follows: capillary voltage, 3kV in negative ion mode and positive ion mode; cone voltage, 40 V; ion source temperature, 120 °C; desolvation temperature, 450 °C; desolvation gas (N_2_) flow rate, 750 L/h; the first range scan, *m*/*z* 100–1600 Da; collision gas, Argon. During low energy scanning, trap collision energy was 4 eV, transfer collision energy was 6 eV, during high energy scanning, trap collision energy was 15 eV, transfer collision energy was 30–50 eV. The mass range was from *m*/*z* 50 to 1500. Leucine-enkephalin (*m*/*z* 556.2771(+)/554.2615(−)) was selected as the lock mass at a concentration of 400 µg/L and flow rate of 5 µL/min. 

### 3.3. Plant Materials

The *Russula vinosa* Lindbl was collected from Jinggangshan (about 800 meters above sea level), Jiangxi province, China. The plant material was authenticated by Associate Professor Chunhua Wang of College of Pharmaceutical Engineering of Traditional Chinese Medicine, Tianjin University of Traditional Chinese Medicine and deposited in Tianjin International Joint Academy of Biotechnology and Medicine (No. 20151002CH). 

### 3.4. Extraction and Isolation

The dried fruiting bodies of *R. vinosa* (5 kg) were crushed and soaked with water to remove sugar. Then, the residues were extracted with 95% aqueous ethanol solution (*v/v*) for 5 h, filtered, and the residues were extracted for four times totally according to the above method. The extracting solution was merged for rotatory evaporation until no alcohol taste was present, and then the total liquid extract (243 g) was obtained. The total liquid extracts were chromatographed over silica gel and eluted with methanol in dichloromethane (0–100%, stepwise), yielding nine fractions (Fr.1–Fr.9), respectively. The crude fraction Fr.3 (39.0 g) was further purified by silica gel (v/v, petroleum ether:ethyl acetate, 0–100%, stepwise) to produce two fractions (Fr.3A,3B). Fr.3A (4.0 g) was purified by silica gel (*v*/*v*, petroleum ether:ethyl acetate, 0–100%, stepwise) and washed crystal by methanol to yield **9** (6.0 mg) and **13** (9.0 mg). Fr.3B (5.0 g) was purified by Pr-HPLC (*v/v*, methanol:water) to give **4** (3.0 mg) and **11** (7.0 mg). Fr·4 was chromatographed on a silica gel (*v/v*, petroleum ether:ethyl acetate, 0–100%, stepwise) to give three fractions (Fr.4A–4C). Then, Fr.4A (6.0 g) was further purified by silica gel (petroleum ether:ethyl acetate, 50:1 to 10:1, *v/v*) to produce two fractions (Fr.4A1,4A2). Fr.4A1 was purified by CC over Sephadex LH-20 gel eluted with CH_2_Cl_2_/MeOH (1:1, *v/v*) to yield **10** (4.0 mg) and **12** (10.0 mg). Fr.4A2 was subjected to CC over silica gel, eluted with petroleum ether:ethyl acetate (100:1, *v/v*), and preparative TLC to yield **2** (5.0 mg) and **15** (3.0 mg). Fr.4B (7.0 g) was purified by silica gel (*v/v*, petroleum ether:ethyl acetate, 0–100%, stepwise) to give **8** (6.0 mg), **14** (7.0 mg), Fr.4B2, and other fractions. Fr.4B2 was purified by CC over Sephadex LH-20 gel eluted with CH_2_Cl_2_/MeOH (2:1, *v/v*), TLC (petroleum ether:ethyl acetate, 4:1, *v/v*) to yield **1** (4.0 mg). Fr.5 (7.0 g) was sent to a silica gel (*v/v*, dichloromethane:methanol, 0–100%, stepwise) to get **6** (3.0 mg), **7** (10.0 mg), and Fr.5-C. Fr.5-C was purified by Pr-HPLC eluted with methanol/water to offer **3** (6.0 mg) and **5** (4.0 mg). See Figure 6. 

#### 3.4.1. Vinosane (**1**)

Colorless needle crystal; [α]${}_{D}^{20}$ −37.8 (CH_3_OH, *c* 0.10); ^1^H- and ^13^C-NMR data see Table 1 and Table 2; HR-ESI-MS *m/z* 249.1503 [M − H]^−^; (calcd for C_15_H_22_O_3_, 249.1491).

#### 3.4.2. Rulepidadione C (**2**)

White amorphous powder; [α]${}_{D}^{20}$ −30.5 (CH_3_OH, *c* 0.13); ^1^H- and ^13^C-NMR data see Table 1 and Table 2; HR-ESI-MS *m/z* 251.1651 [M + H]^+^; (calcd for: C_15_H_22_O_3_, 251.1647).

#### 3.4.3. (24*E*)-3,4-Seco-cucurbita-4,24-diene-26,29-dioic acid-3-methyl ester (**3**)

White amorphous powder; [α]${}_{D}^{20}$ +47.8 (CH_3_OH; *c* 0.34); ^1^H- and ^13^C-NMR data see Table 1 and Table 2; HR-ESI-MS *m/z* 515.3375 [M − H]^−^; (calcd for: C_31_H_48_O_6_, 515.3373).

#### 3.4.4. (24*E*)-3,4-Seco-cucurbita-4,24-diene-26-oic acid-3-ethyl ester (**4**)

White amorphous powder; [α]${}_{D}^{20}$ +13.5 (CH_3_OH, *c* 0.21); ^1^H- and ^13^C-NMR data see Table 1 and Table 2; HR-ESI-MS *m/z* 523.3405 [M + Na]^+^ (calcd for: C_32_H_52_O_4_, 523.3763).

#### 3.4.5. (24*E*)-3β-Hydroxycucurbita-5,24-diene-26,29-dioic acid (**5**)

White amorphous powder; ^1^H- and ^13^C-NMR data see Table 1 and Table 2; HR-ESI-MS *m/z* 485.3275 [M − H]^−^ (calcd for: C_30_H_46_O_5_, 485.3267).

#### 3.4.6. (2*S*,3*S*,4*R*,2′*R*)-2-(2′-Hydroxydocosanoylamino) eicosane-1,3,4-triol (**6**)

White granular powder; ^1^H- and ^13^C-NMR data see Table 3; LC-IT-TOF-MS *m/z* 684.6488 [M + H]^+^ (calcd for: C_42_H_85_NO_5_, 684.6506).

### 3.5. Biological Activity Assays

Cell culture, CCK-8 cell viability assay, anti-inflammatory activity assay, and statistical analysis were carried out according to the procedure we published [20,21].

## 4. Conclusions

*R. vinosa*, a multifunctional edible mushroom, has been used to make delicious food or soup in the south of China, but its nutritional ingredients are not clear. Therefore, we studied its chemical constituents with high interest. Through the isolation and purification of the compounds in *R. vinosa*, 15 compounds, including six new ones, were obtained. We used NMR, MS, and polarimetry to identify the chemical structures of the new compounds. From the isolated compounds, we could see that triterpenoids and sesquiterpenoids were its main chemical constituents, which was similar to other mushrooms in this genus. In the previous report, illudoid sesquiterpenes were obtained from the fruit body of *R. japonica* with neurite outgrowth promoting activity [8]; triterpenes and sesquiterpenes were found from *R. lepida* with anti-tumor activity [10]; the chemical composition was researched on *R. griseocarnosa* sp. nov. with antioxidant activities [22], etc. Thus, our work on the chemical constituents of *R. vinosa* could give a chemical material base of this fungus and enrich the chemical constituents of this genus. After the isolation and structure elucidation, the bioactivity of these compounds was evaluated. Referring to the activities of the components obtained from this genus, inhibitory activity on NO production was selected to assay their effects. Therefore, firstly, the cytotoxic evaluation on compounds **1**–**5** and **7**–**13** against mouse macrophages RAW 264.7 cells showed that they had little cytotoxic activity. Then, inhibitory activity of these compounds on NO production was performed. As a result, compounds **1**, **3**, **4**, **7**, **12**, and **13** showed moderate activity of inhibiting the LPS, inducing cells to produce NO at higher concentrations. This result suggests that *R. vinosa* has potential anti-inflammatory effect. From the marked peaks in the MS chromatograms, we can see that there are many other compounds that we have not obtained yet, so more investigation on its chemistry should be done in the future. For the bioactivity of these isolates, the inhibitory effect on NO production was only chosen in this work, and other evaluations, such as anti-oxidant, anti-bacterial, and immune enhancement activities should be done in the future.

## Figures and Tables

**Figure 1 molecules-24-01305-f001:**
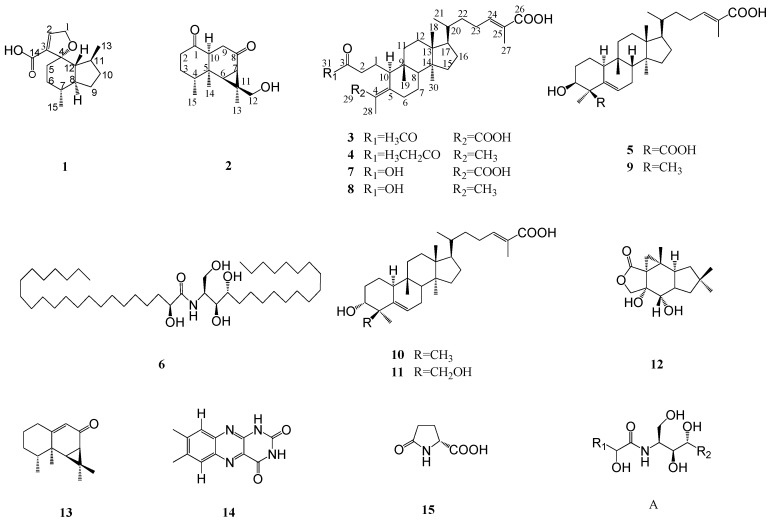
Chemical structures of compounds **1**–**15** and core structure of compound **6** (**A**).

**Figure 2 molecules-24-01305-f002:**
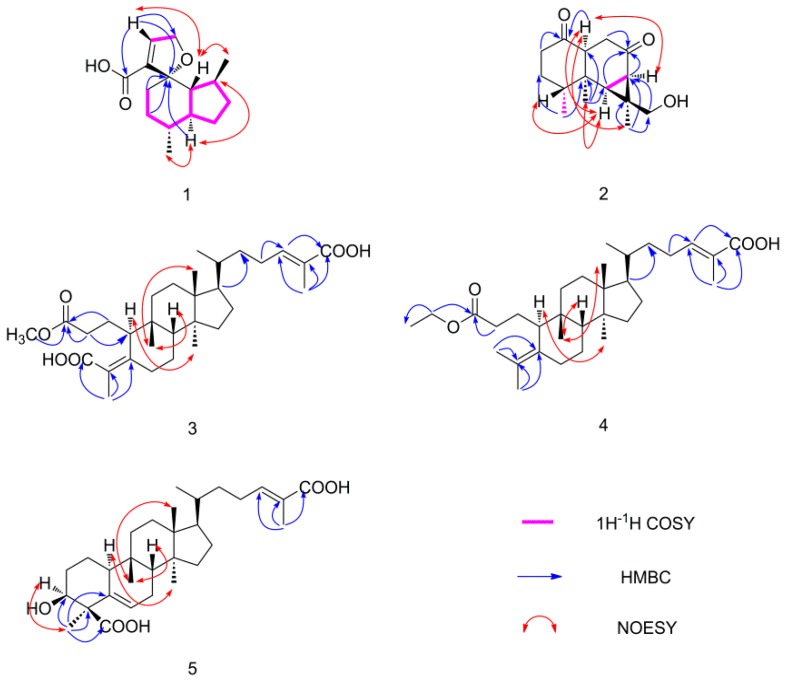
Key ^1^H-^1^H COSY, HMBC, NOESY correlations of compounds **1**–**5.**

**Figure 3 molecules-24-01305-f003:**
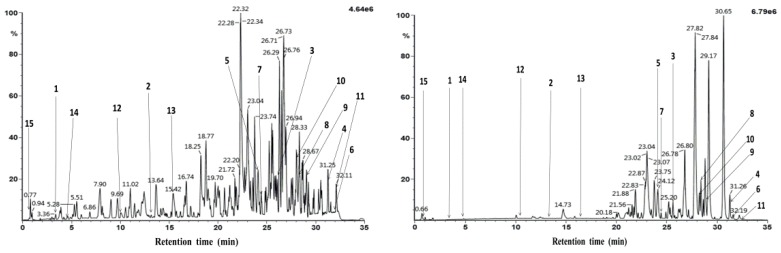
Positive and negative ion mode chromatograms of 95% ethanol extract from the *R. vinosa* with marked peaks of isolated compounds.

**Figure 4 molecules-24-01305-f004:**
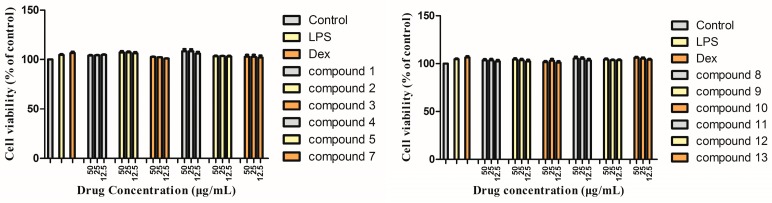
The effects of the isolated compounds on the survival rate of LPS-induced RAW264.7 macrophage cells. The cells were pre-treated with concentrations (50, 25, 12.5 μg/mL) of compounds (**1**–**5**, **7**–**1****3**) or dexamethasone sodium phosphate (Dex) (5 μg/mL) for 1 h, then stimulated with LPS (0.1μg/mL) for 16 h. The data show the mean ± SD of three independent experiments performed in triplicates.

**Figure 5 molecules-24-01305-f005:**
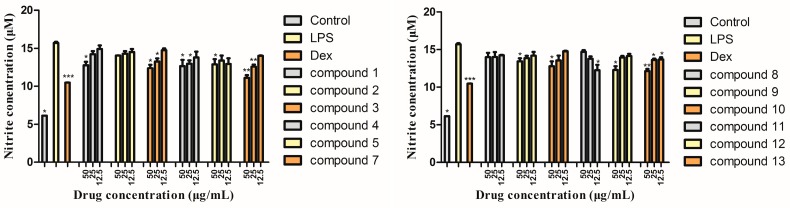
Inhibitory effects on NO production of the isolated compounds. The cells were pre-treated with concentrations (50, 25, 12.5 μg/mL) of compounds (**1**–**5**, **7**–**1****3**) or dexamethasone sodium phosphate (Dex) (5 μg/mL) for 1 h, then stimulated with LPS (0.1μg/mL) for 16 h. The data show the mean ± SD of three independent experiments performed in triplicates. * *p* < 0.05, ** *p* < 0.01, *** *p* < 0.001, compared to the LPS-treated values.

**Figure 6 molecules-24-01305-f006:**
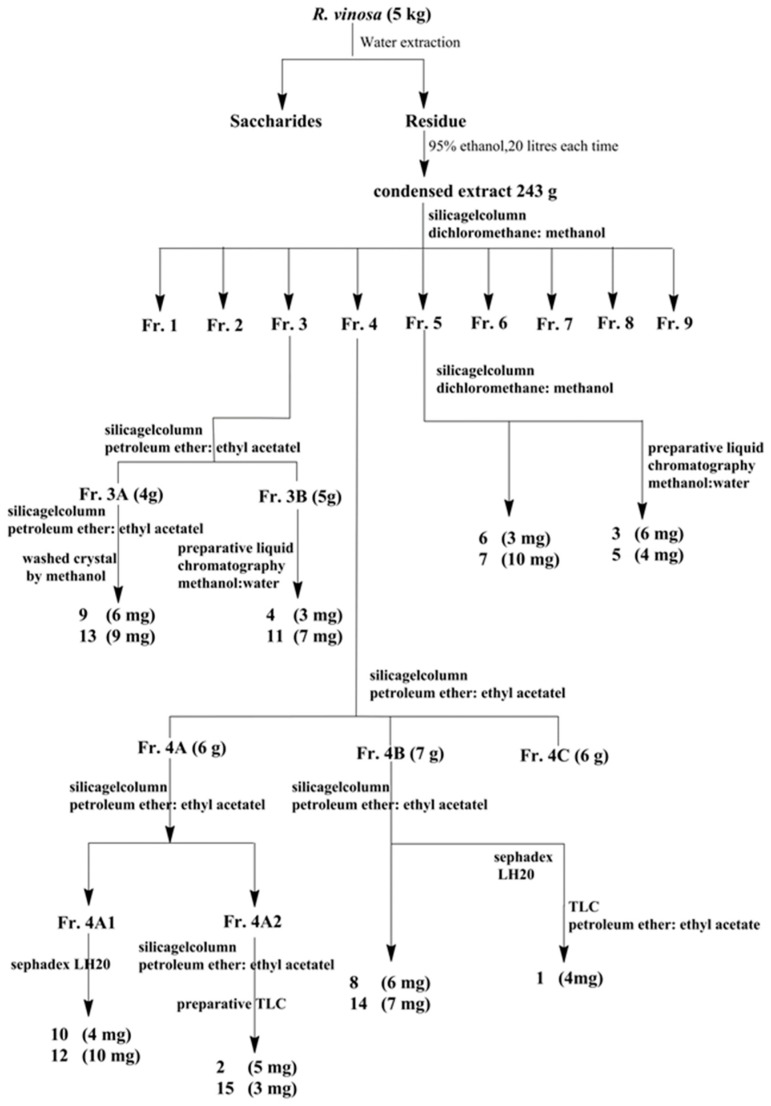
Scheme illustrating the entire isolation process.

**Table 1 molecules-24-01305-t001:** ^1^H-NMR data for compounds **1**–**5** (400 MHz, δ in ppm, *J* in Hz).

No.	1 ^a^	2 ^b^	3 ^a^	4 ^a^	5 ^c^
1	4.78 (m)		1.68 (m), 1.95 (m)	1.26 (m), 1.85 (m)	1.60 (m)
2	6.82 (s)	2.33 (m), 2.58 (m)	2.31 (m)	2.25 (m)	2.03 (m)
3		1.75 (m), 2.02 (m)			4.23 (m)
4		2.27 (m)			
5	1.56 (m), 1.34 (m)				
6	1.28 (m), 1.53 (m)	1.49 (d, 8.1)	2.14 (m), 3.13 (m)	2.00 (m), 2.42 (m)	6.01 (d,5.6)
7	1.75 (m)	1.75 (m)	1.24 (m), 1.85 (m)	1.39 (m), 1.93 m)	2.31 (m), 1.75 (m)
8	2.05 (m)		2.48 (dd, 7.5, 5.7)	1.93 (m)	1.67 (m)
9	1.61 (m), 1.48 (m)	2.16 (m), 2.49 (m)			
10	1.48 (m)	3.08 (dd, 12.6, 5.2)	1.96 (1H, m)	2.42 (m)	2.44 (m)
11	2.12 (m)		1.50 (m)	1.58 (m), 1.71 (m)	1.67 (m), 1.39 (m)
12	1.86 (t, 6.0)	3.37 (m)	1.25 (m), 1.50 (m)	1.12 (m)	1.47 (m)
13	0.96 (d, 7.2)	1.42 (s)			
14		0.88 (s)			
15	0.90 (d, 6.8)	1.15 (d, 6.8)	1.13 (m)	1.49 (m), 1.70 (m)	1.61 (m), 1.45 (m)
16			1.50 (m), 1.98 (m)	1.56 (m), 1.86 (m)	1.84 (m), 1.20 (m)
17			1.49 (m)	1.52 (m)	1.47 (m)
18			0.82 (s)	0.81 (s)	0.81 (s)
19			1.19 (s)	1.14 (s)	0.97 (s)
20			1.44 (m)	1.46 (m)	1.48 (m)
21			0.91 (d, 6.0)	0.91 (d,6.0)	0.97 (s)
22			1.50 (m)	1.19 (m), 1.55 (m)	1.05 (m), 1.15 (m)
23			2.24 (m), 2.09 (m)	2.12 (m), 2.26 (m)	2.39 (m), 2.15 (m)
24			6.90 (m)	6.90 (s)	7.23 (m)
25					
26					
27			1.84 (s)	1.83 (s)	2.11 (s)
28			1.84 (s)	1.64 (s)	1.66 (s)
29				1.54 (s)	
30			0.66 (3H, s)	0.63 (3H, s)	0.90 (s)
OCH_3(2)_			3.66 (3H, s)	4.13 (2H, m)	
CH_3_				1.25 (3H, t, 7.1)	

^a^ In chloroform-*d*_1_.^b^ In methanol-*d*_4_.^c^ In pyridine-*d*_5_.

**Table 2 molecules-24-01305-t002:** ^13^C-NMR Data (δ in ppm) for compounds **1**–**5** (100 MHz).

No.	1 ^a^	2 ^b^	3 ^a^	4 ^a^	5 ^c^
1	72.2	211.8	26.2	27.9	21.5
2	129.5	41.5	32.5	33.1	29.6
3	152.3	32.0	174.4	174.7	72.9
4	92.6	40.5	121.9	123.2	55.0
5	37.3	41.0	154.7	133.8	138.8
6	24.1	35.4	24.6	22.4	123.3
7	32.4	33.8	27.9	22.5	24.9
8	46.1	211.2	43.8	43.9	44.2
9	23.4	35.2	38.1	37.5	35.4
10	31.8	51.4	43.6	42.8	37.7
11	39.3	33.8	36.9	36.8	32.7
12	49.0	72.0	30.0	34.4	35.9
13	17.2	13.8	45.8	45.9	46.9
14	168.0	16.0	48.9	49.2	49.9
15	20.1	15.5	34.2	30.3	31.1
16			23.0	26.6	28.5
17			51.0	51.0	51.1
18			15.4	15.6	15.9
19			30.7	30.8	28.2
20			36.3	36.2	36.5
21			18.6	18.6	19.1
22			35.1	35.0	35.2
23			26.0	26.0	26.3
24			146.0	146.0	142.9
25			126.8	126.7	129.3
26			173.8	173.2	171.0
27			12.0	12.1	13.2
28			16.5	20.4	23.1
29			175.8	21.4	180.5
30			17.4	17.4	18.2
OCH_3(2)_			51.7	60.3	
CH_3_				14.4	

^a^ In chloroform-*d*_1_.^b^ In methanol-*d*_4_.^c^ In pyridine-*d*_5_.

**Table 3 molecules-24-01305-t003:** ^1^H-NMR and ^13^C-NMR spectral data of compound **6** (400 MHz and 100 MHz, pyridine-*d*_5_, *J* in Hz).

Position	^13^C	^1^H
1	62.4	4.53 (m), 4.38 (m)
2	53.4	4.45 (dd, 10.8, 4.5)
3	77.2	5.14 (m)
4	73.4	4.31 (m)
5	34.5	2.27 (m), 1.95 (m)
6	27.0	1.20–1.85 (overlapped)
7–18	29.5–32.5 (overlapped)	1.20–1.85 (overlapped)
19	23.3	1.20–1.85 (overlapped)
20	14.7	0.89 (m)
1′	175.6	
2′	72.9	4.64 (dd, 10.8, 4.5)
3′	36.1	2.24 (m), 2.07 (m)
4′	26.2	1.20–1.85 (overlapped)
5′–20′	29.5–32.5 (overlapped)	1.20–1.85 (overlapped)
21′	23.3	1.20–1.85 (overlapped)
22′	14.7	0.89 (m)

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
