# Peer review of "Chemical Constituents with Inhibitory Activity of NO Production from a Wild Edible Mushroom, Russula vinosa Lindbl, May Be Its Nutritional Ingredients"

_molecules, 2019, doi:10.3390/molecules24071305_

Round 1
Reviewer 1 Report
This submission investigated on the chemical constituents of an edible mushroom named Russula vinosa Lindbl. I like to give the following comments.
1. The fat-soluble components were analyzed without rationale introduction.
2. How to prepare each component in solution for assay of bioactivity? It was not indicated in clear.
3. Cytotoxic activity was negative. Why? It seems not associated with the anti-tumor effects as described previously. Please give a suitable reason in discussion.
4. In Figure 3, DEX means what?
5. LPS-induced NO production in RAW 264.7 mouse macrophages as the inflammatory indicator is only one part but not the real anti-inflammation.
6. Variation in ED50 between each component is required. Which one belonged to the most effective component?
7. In conclusion, compounds 1-5, 7-13 showed cytotoxic activity but it was not observed in Figure 3.
8. Limitation(s) of this report shall be concerned in detail.
Author Response
Dear Editor
On behalf of the co-authors, I appreciate the opportunity to submit the revised manuscript in response to the reviewers’ suggestions. Meanwhile, I would also like to express our sincere appreciation to the reviewers because their rigorous and constructive comments helped us tremendously to improve the quality and the potential impact of this manuscript.
We have addressed the comments fully one by one following each comment in the following section. We believe the revision has addressed reviewer’s comments and greatly helped us improving the manuscript.
According to the comments, we have revised and highlighted the relevant part in the revised manuscript.
Reviewer 1.
This submission investigated on the chemical constituents of an edible mushroom named Russula vinosa Lindbl. I like to give the following comments.
1. The fat-soluble components were analyzed without rationale introduction.
Respond:
We have added several reported bioactivities of fat-soluble components from mushroom in the revised version. The fat-soluble components have some good bioactivities, so, we had searched this kind of compounds to find new leading compounds. So, we added some literatures in the introduction part.
2. How to prepare each component in solution for assay of bioactivity? It was not indicated in clear.
Respond:
The method of assay was same as the work we have published, in order to reduce the length of the article, we gave the references we previously published.
Cell Culture, CCK-8 cell Viability Assay, Anti-Inflammatory Activity Assay, Statistical Analysis, were carried out according to the procedure we published [20,21].
3. Cytotoxic activity was negative. Why? It seems not associated with the anti-tumor effects as described previously. Please give a suitable reason in discussion.
Respond:
Before evaluating the NO inhibitory activity of these compounds, we need to evaluate the cytotoxicity of these candidate compounds to cells. If the compounds have no cytotoxic activity to cells at the corresponding concentration, it is meaningful for us to do NO inhibitory activity.
So, our aim is not to evaluate the cytotoxicity of these compounds.
4. In Figure 3, DEX means what?
Respond:
We have given the full name, Dexamethasone Sodium Phosphate.
5. LPS-induced NO production in RAW 264.7 mouse macrophages as the inflammatory indicator is only one part but not the real anti-inflammation.
Respond:
Thank you for your good comment, and I don't think this title is suitable, either. So, we changed the title as follow:
Chemical constituents with inhibitory activity of NO production from a wild edible mushroom, Russula vinosa Lindbl, may be its nutritional ingredients.
6. Variation in ED50 between each component is required. Which one belonged to the most effective component?
Respond:
In this work, we wanted to preliminarily evaluate the NO inhibitory activity of these compounds. From the histogram, we can clearly see that compound 7 has significant NO inhibitory activity at 50 μg/mL concentration.
7. In conclusion, compounds 1-5, 7-13 showed cytotoxic activity but it was not observed in Figure 3.
Respond:
“The cytotoxic evaluation on compounds 1-5, 7-13 against mouse macrophages RAW 264.7 cells showed that they had little cytotoxic activity.”
In other words, these compounds had no obviously cytotoxicity on the cells at the experimental concentrations.
8. Limitation(s) of this report shall be concerned in detail.
Respond:
Thanks for this suggestion; we have added the limitations in the conclusion part.

Reviewer 2 Report
The aim of the paper was to isolate the chemical constituents from wild edible mushroom Russula vinosa Lindbl, and determinate their cytotoxicity by Cell Counting Kit-8 and inhibitory effects on Nitric Oxide (NO) production in LPS-stimulated mouse macrophage RAW 264.7 cells. The idea of the work is new, since fifteen compounds, including six new ones, were isolated and their full structure was elucidated. The experimental part lacks, however, numerous data important for repetition of the work and verification of its scientific soundness. The work lacks also proper discussion - the obtained results are simply described with no mention of any possible implications and impact of the experimental observations. In my opinion, the submission requires extensive edition and improvement at many points, main of which are listed below.
Introduction:
L40-58: This whole paragraph should be transferred to the Results and Discussion section.
The Authors should describe in the Introduction in more detail the phytochemical profile of the Russula vinosa species compared to other representatives of the Russula genus. In the next part of this paragraph, the Authors should explain why such biological model, i.e. inhibitory effect of isolated compounds on nitric oxide production in LPS-activated macrophages RAW 264.7, was chosen for research and if there are any literature reasons for it.
Results and Discussion
2.1. Structural elucidation
L63: The Authors should add a LC-MS chromatogram of 95% ethanol extract from the R. vinosa with marked peaks of isolated compounds.
L94: Please replace Figure 2 with a better resolution figure.
L116: should be HR-ESI-MS. Please check the whole manuscript thoroughly.
L185: Please add the structure as a separate figure.
2.2. Bioactivity evaluation
L202 and 212: Figure 3 and 4 are not very readable. Please prepare new figures, that are definitely bigger, maybe colorful, but certainly with larger gaps between the bars for individual compounds.
The work lacks proper discussion - the obtained results are simply described with no mention of any possible implications and impact of the experimental observations. In this section the critical discussion with broader adequate literature background is necessary. The biological activity of nine known compounds (7-15), isolated from Russula vinosa, should be discussed and compared with the data received by other Authors.
Materials and Methods
3.1. Plant materials
L220: The Authors should add collection site of the plant material, alt. (m), and coordinates.
L226: should be v/v (written in italics). Please check the whole manuscript thoroughly.
L231-245: The Authors should add a scheme illustrating the entire isolation process (solvents, fractions, isolated compounds and their yields).
The Authors should add a paragraph concerning general experimental procedures, i.e. preparative column chromatography, preparative HPLC, preparative TLC, and equipment used in HR-ESI-MS, and NMR analyses.
Author Response
Reviewer 2.
1. The aim of the paper was to isolate the chemical constituents from wild edible mushroom Russula vinosa Lindbl, and determinate their cytotoxicity by Cell Counting Kit-8 and inhibitory effects on Nitric Oxide (NO) production in LPS-stimulated mouse macrophage RAW 264.7 cells. The idea of the work is new, since fifteen compounds, including six new ones, were isolated and their full structure was elucidated. The experimental part lacks, however, numerous data important for repetition of the work and verification of its scientific soundness. The work lacks also proper discussion - the obtained results are simply described with no mention of any possible implications and impact of the experimental observations. In my opinion, the submission requires extensive edition and improvement at many points, main of which are listed below.
Respond:
As your comments, we have given some discussions in the conclusion part.
1. Introduction:
L40-58: This whole paragraph should be transferred to the Results and Discussion section.
Respond:
We have transferred them to the Results and Discussion section.
The Authors should describe in the Introduction in more detail the phytochemical profile of the Russula vinosa species compared to other representatives of the Russula genus. In the next part of this paragraph, the Authors should explain why such biological model, i.e. inhibitory effect of isolated compounds on nitric oxide production in LPS-activated macrophages RAW 264.7, was chosen for research and if there are any literature reasons for it.
Respond:
Thanks for your good suggestion. We re-read several papers about the chemical constituents of this genus. And, in the introduction part, we added some published work about the chemical constituents about this genus. Several triterpenes and sesquiterpenes have been obtained from this genus, which have anti-tumor, ant-inflamation activity, etc. To preliminarily evaluate the activity of these isolate, the inhibitory effect on NO production of these compounds was chosen in our work. As we know, this evaluation does not fully evaluate the activity of these compounds, so, other activity assays will be done in the future.
2. Results and Discussion
2.1. Structural elucidation
L63: The Authors should add a LC-MS chromatogram of 95% ethanol extract from the R. vinosawith marked peaks of isolated compounds.
Respond:
Thank you for your good suggestion, we supplemented this experiment, and the 15 compounds were all marked in the MS chromatogram according to their MWs. So, we added the LC-MS chromatograms in the revision, and the identification of these 15 isolates in the SI file (Tables S1 and S2).
3. L94: Please replace Figure 2 with a better resolution figure.
Respond:
We have replaced the better figure as your suggestion.
4. L116: should be HR-ESI-MS. Please check the whole manuscript thoroughly.
L185: Please add the structure as a separate figure.
Respond:
Ok, we changed them to HR-ESI-MS.
We added the structure in Line 185 in Figure 1(A).
5. Bioactivity evaluation
L202 and 212: Figure 3 and 4 are not very readable. Please prepare new figures, that are definitely bigger, maybe colorful, but certainly with larger gaps between the bars for individual compounds.
Respond:
Thank you, we changed the Figures in the revision.
6. The work lacks proper discussion - the obtained results are simply described with no mention of any possible implications and impact of the experimental observations. In this section the critical discussion with broader adequate literature background is necessary. The biological activity of nine known compounds (7-15), isolated from Russula vinosa, should be discussed and compared with the data received by other Authors.
Respond:
Thanks for this, we added more discussions as you suggested in the conclusion part.
7. Materials and Methods
3.1. Plant materials
L220: The Authors should add collection site of the plant material, alt. (m), and coordinates.
Respond:
The plant materials were collected in Jinggang Mountain (Jinggangshan) of Jiangxi province, China. So, in this revised version, we give the approximate height.
L226: should be v/v (written in italics). Please check the whole manuscript thoroughly.
Respond:
Thank you, I have revised this.
L231-245: The Authors should add a scheme illustrating the entire isolation process (solvents, fractions, isolated compounds and their yields).
Respond:
In the revised version, we give the scheme.
The Authors should add a paragraph concerning general experimental procedures, i.e. preparative column chromatography, preparative HPLC, preparative TLC, and equipment used in HR-ESI-MS, and NMR analyses.
Respond:
Ok, we have added this part.
Round 2
Reviewer 2 Report
The manuscript has been revised in accordance with the Reviewers' recommendations and is now suitable for publication in Molecules.